# Does multifamily therapy help parents of adolescents with anxiety-based school refusal? A qualitative approach

Claire Snegaroff[1], Maude Ludot-Grégoire[1,2,3], Sara Skandrani[4], Caroline Barry[3], Christine Hassler[3], Salomé Grandclerc[1,2,3], Sevan Minassian[1,2,3], Eloïse Hellier[1,2,3], Rahmeth Radjack[1,2,3], Marie Rose Moro[1,2,3], Aurélie Harf[1,2,3]*

1 Maison de Solenn-Maison des Adolescents, Cochin Hospital, AP HP, Paris, France, 2 Université Paris Cité, Paris, France, 3 INSERM, CESP, Villejuif, France, 4 Université Paris Nanterre, Nanterre, France

* aurelie.harf@aphp.fr

## Abstract

### Introduction

Anxiety-based school refusal is a major public health problem among adolescents, with serious professional, social, and psychiatric consequences in adulthood. Family and especially parental involvement is essential. Multifamily therapy has been shown to be effective in the management of many psychological disorders including anxiety-related disorders. The aim of this qualitative study is to explore the experiences of parents who participated in the MULTI FAST program (a multifamily therapy focused on anxiety-based school refusal).

### Method

Participation in the study was offered to all families who participated in a MULTI FAST group in 2022 and 2023. A semi structured interview was conducted with each parent who agreed to participate, six months after the end of the multi-family therapy. The interviews were analyzed with a qualitative phenomenological method, *Interpretative Phenomenological Analysis*.

### Results

Thirty-one parents (out of 37, 84%) agreed to take part in the study: 19 mothers and 12 fathers. The mean age for fathers was 51.5 (SD = 6.45) and 48.7 (SD = 4.77) for mothers. Analysis of the interviews revealed four main themes. The first was changes in parents' representations of their adolescent, with a better understanding of their child's distress and needs. The second theme was parents looking at themselves in a new light, with less sense of abnormality, less guilt, and looking back at their own relation to school. The third theme was the expression of parental distress to their adolescents, and the fourth described better communication and stronger family groups after the multifamily therapy.

**Data availability statement:** All relevant data are within the manuscript and its Supporting information files.

**Funding:** The author(s) received no specific funding for this work.

**Competing interests:** The authors have declared that no competing interests exist.

**Abbreviations:** ABSR, anxiety-based school refusal; MFT, Multifamily therapy; DSM, Diagnostic and statistical manual; COREQ, Consolidated criteria for Reporting Qualitative research; IPA, Interpretative Phenomenological Analysis; AP-HP, Assistance Publique–Hôpitaux de Paris.

## Discussion

All parents who participated in the study emphasized the help that multifamily therapy had given them. This included an increased sense of parental self-efficacy, better mentalization, and a positive impact on family interactions, with changes in everyone's place in the family.

## Introduction

### What is school refusal?

School refusal is a real public health issue in many countries. Absenteeism from school has repercussions on future educational and professional plans [1,2] and increases the child's risk of dropping out of school. School refusal can involve a variety of situations [3–5]. In this paper we focus on one type of school attendance problem: school refusal, which is what we encounter most frequently in our adolescent psychiatry department. Anxiety, excessive fearfulness, unhappiness, unexplained physical symptoms, depressive affect, and sleep problems are all described with school refusal [6]. Parents are aware of these children's inability to attend class and have tried several strategies to help them return to school [3].

These adolescents also meet Berg's five criteria for school refusal [7]: (a) reluctance or refusal to attend school; (b) being at home during school hours with parental knowledge; (c) emotional distress at the prospect of school; (d) an absence of severe antisocial behaviors beyond resistance to parental attempts to enforce school attendance; and (e) reasonable yet ineffective attempts by parents to enforce this.

The association between school refusal and anxiety [8] (school anxiety, separation anxiety), in particular, has been noted by others, as highlighted by Tekin and Aydin in their review of 30 research articles [9].

Epidemiological studies report that school refusal occurs among 1–7% of youths in the general population and 5–16% of youths referred to clinical settings [10]. Looking more specifically at adolescent studies, Havik et al. found that 3.6% of 11–15-year-olds reported "quite often" on at least one item included in the school refusal related reasons for school non-attendance in the three months before the interview [11]. That study interviewed 5465 students aged from 11 to 15 years old. Their criteria are therefore different from Berg's criteria defining school refusal. In the community sample of 11–17-year-olds studied by Steinhausen et al., 6.9% reported fear of going to school [12]. No specific gender differences were found for the duration of school absence [13]. These data show the difficulty in obtaining rigorous epidemiological data on school refusal, as the criteria and the populations vary widely from one study to another.

### What's special about school refusal in adolescence?

Adolescents have a higher incidence of anxiety-based school refusal (ABSR) than children. Burnham et al. reported that U.S. youths aged 12–19 years were twice as likely to demonstrate a profile of school-related fears than youths aged 7–11 years [14].

Several authors describe more severe forms of ABSR in adolescents compared with children and a greater presence of associated anxiety disorders, particularly social anxiety [15]. In one study, more than two-thirds of adolescents with school refusal (12–18 years) were diagnosed with social anxiety disorder [16]. Another study highlights the very high prevalence of social anxiety in adolescents (11–19 years) with ABSR [17]. This social anxiety in adolescence inevitably has consequences both for schooling (avoidance of anxiety-provoking situations, lack of class participation, grades below the student's level) and relationships with others (e.g., isolation, loneliness, and lack of group support) [15].

A retrospective study found that almost half of a sample of adults with anxiety disorders reported having left school prematurely [18]. The reasons given indicated that 22.4% experienced anxiety (felt nervous in class or at school) and 16.9% showed behaviors related to anxiety (problems participating in class).

The question of bullying must also be addressed. Indeed, in France, nearly half of parents whose teenager has ABSR report that their child has been bullied at school [19]. The correlation between bullying and school refusal must be considered in caring for both children and adolescents with school refusal.

The picture in adolescence is therefore complex, and treatment must take into account the severity of the school refusal, associated anxiety and depressive disorders, and peer relationships.

## Working with the family

A first reason to offer family interventions is that systemic family therapy has been shown to be effective in adolescents' psychiatric disorders. A systematic review of 85 randomized trials showed its efficacy for internalizing and externalizing disorders of childhood and adolescence [20,21]. Family therapies have also been shown to be effective specifically for anxiety disorders [22].

The second reason for family treatment is that anxiety levels are higher among parents whose teenager has ABSR. Having a child who is no longer able to go to school, trying everything to help him or her get back in, and failing — this experience induces substantial exhaustion, guilt, and anxiety. School refusal can contribute to parental frustration and helplessness [23] and cause family stress and conflict [24].

Parents of adolescents with ABSR have a higher prevalence of mental health problems than parents in the general population [13]. McShane et al. found a high prevalence of mental health problems among mothers (53%) and fathers (34%) of youths with anxiety based school attendance problems [25]. Parental depression and parental anxiety are strong predictors of school refusal [9].

Finally, the third reason for working with families is that a significant proportion of teenagers do not respond to current treatments, especially individual, for ABSR. The authors explain that parents, even when they play a key role, are not sufficiently involved: it is essential to take parents' psychopathology and family functioning into account [26]. Similarly, Gonzalvez stresses the importance of working on family functioning to help adolescents with this form of school refusal [15]. Maynard et al. reviewed 8 studies describing different treatments for school refusal among children and adolescents; 6 of them actively involved parents [27]. These articles show the essential role that families play in the treatment of adolescents with school refusal.

Currently cognitive-behavioral therapy has the most scientific support in the treatment of school refusal [27–29]. Heyne points out that among adolescents, cognitive-behavioral therapy fails in one- to two-thirds of cases, although it is more effective for children than for adolescents [30]. He suggests more intensive treatment and greater family involvement. Seven points emerge from this brief literature survey: more frequent sessions or simply more sessions, greater attention to social anxiety disorder and social functioning, careful consideration of parents' roles, greater attention to both parent-adolescent communication and problem-solving, and the use of alternative educational settings.

There are very few studies on the effectiveness of drug treatments with antidepressants or benzodiazepines, and their efficacy is low. These drugs are mostly used to treat comorbidities [31].

The management of ABSR in adolescence must therefore involve the family as a central partner in care. It must also work on social anxiety and, consequently, relational difficulties with other adolescents.

## Multifamily therapy

Multifamily therapy (MFT) meets several of the criteria suggested by the literature described above: involving families from the outset, meeting other families facing the same problem, school-refusing teens spending time with other teenagers, and exposure to group situations. MFT brings together several families facing the same problem, be it psychological, somatic, social, or an interweaving of several of these dimensions. Briefly defined, group therapy + family therapy = MFT.

MFT has developed since the 1960s, initially in the USA, with chronically psychotic patients and their families [32]. From the outset, caregivers were struck by the benefits of this group approach, particularly in terms of alleviating the sense of isolation and stigmatization felt by families, who often described being misunderstood or judged by those around them. In the 1980s, McFarlane — also in the USA and working with families with one member who had schizophrenia — described the three axes of the psycho-educational dimension of MFT: the pedagogical axis, aimed at better explaining the illness and treatments, the psychological axis, which tackles issues linked to living with a person with schizophrenia, and the behavioral axis with communication models and solving everyday problems [33].

MFT developed all around the world, for multiple issues and disorders [34]. Among the studies showing the effectiveness of MFT, several include adolescents, in particular, for eating disorders [35], autism spectrum disorders [36], attention deficit hyperactivity disorder [37], conduct disorder [38], post-traumatic stress disorder [39], internet addiction [40], and drug abuse [41]. In the 2023 systematic review and meta-analysis of MFT effectiveness by Van Es et al., 55% of the studies reviewed included children or adolescents [42].

At the same time, parent training programs have long been recognized as valuable means of dealing with school-related issues involving children's behavioral, social, and academic problems. Families and Schools Together (FAST), a program in England used in 115 schools since 2018, offers a multifamily support group designed to increase parental involvement in school and improve children's well-being. Also in England, Asen founded the Family School, which uses the same principle to bring parents into the school so that teaching teams and families can work together around children with academic and behavioral difficulties [43].

This literature review thus highlights the effectiveness of MFT, even despite the great heterogeneity of the problems and populations described in the studies above. We show here the components that make it especially valuable for ABSR in adolescence. It enables work with the family: psycho-education, reducing conflict, improving communication, setting up strategies for returning to school or resuming a project. In MFT, to reduce conflict and improve communication, we work on mentalization within families. Mentalization, as described by Fonagy, enables a better understanding of others, their mental states, and behaviors [44]. Putting oneself in the shoes of the person with whom one is interacting makes it possible to apprehend and appreciate the meaning of his or her conduct, emotions, and speech. Many exercises in MFT aim to improve mentalization and, in particular, to enhance the parental reflective function [44], for the recognition of the other's thoughts and feelings improves family communication.

At the same time, the group setting is highly beneficial for these often isolated adolescents, who find themselves in a group with other adolescents who are going through the same thing. It is an equally helpful setting for parents, who very often feel not only isolated and anxious themselves, but also guilty. Participating with other parents in similar situations lightens the parental burden and helps them to break out of their isolation and guilt.

## Objectives

The aim of this paper, after this brief review of school refusal and MFT, is to explore the experiences of parents who participated in one of our MFT groups for ABSR to allow us to examine the participants' experiences and thereby improve the therapy we offer for the adolescents and families we treat. The qualitative method, which relies on analyzing the discourse

of the people involved, putting them in the position of experts, is particularly well-suited to this context and this goal. More specifically, then, we seek to analyze the participants' discourse and thereby gain better insight into the processes leading to change and clinical improvement for patients and their families. In particular, analysis of these interviews has already enabled us to develop new objectives to be worked on during MFT sessions. This article presents the analysis of the parents' discourse. The analysis of adolescent discourse is currently underway and will be presented in a separate article.

## Materials and methods

The description of the methods we used follows the COREQ (Consolidated criteria for Reporting Qualitative research) guidelines [45] (see Additional file).

### The MULTI FAST program

Since 2018, the Maison des Adolescents (Cochin Hospital, Paris, France) has offered MFT to adolescents with ABSR and their families. The program involves 10 sessions of 3 hours each, approximately 3 weeks apart. Each group includes 6–8 families, all consulting for their teenager's ABSR. The duration of school refusal is between 2 weeks and 18 months. Parents and teenagers are invited to participate in all MFT sessions, while siblings are invited to sessions 1, 3, 4, 8, and 10. It is advisable to bring all the people living with the teenager and involved in dealing with the problem, e.g., spouses in the case of parental separation, members of the extended family living in the teenager's home (e.g., grandparents), etc.

Each 3-hour session is divided into two activities, with a break in the middle. Each activity is divided into several parts: an initial period of discussion in subgroups (all teenagers together and parents together, or mothers and fathers separately, or mixed subgroups, or by family), followed by a presentation of the subgroups' discussions, and then by an overall discussion. Depending on the objectives, different activities are proposed. Examples include:

• How to draw school refusal in teen/parent subgroups, followed by discussion between the different groups.

• Animal genograms [46] to describe family members' reactions to the problem, that is, self-description and description of other family members, by using animals: the image of the ostrich evokes avoidance of the problem, the rhinoceros is confrontational, the kangaroo protective or overprotective, the dolphin benevolent, etc. Each person can choose several animals per person and invent other real or imaginary animals).

• The press conference [43] places adolescents as experts on their problem. They take on the role of leading experts on ABSR and respond to journalists at a round-table discussion at an imaginary conference, while parents play the role of journalists and ask the experts questions. This is an emotionally powerful exercise, and one in which parents discover their teenager in a new light. The teenagers are able to express to their parents their convictions on the causes of their school refusal, the factors that help, the evolution, the misconceptions to combat, and so on.

• Everyone sketches his or her vision of school, then there's time for discussion within the family and then between families.

The themes addressed during MFT are:

• Information on ABSR (psycho-education)

• Its causes and triggers

• The various possibilities for returning to school

•  Managing anxiety

• Reflecting on the positive or negative impact of different support strategies

 

- Analysis of how each family member reacts to the problem

- Relationships with peers, extended family, and the family social network

- Projecting into the future

## Population

All included parents participated in a MULTI FAST group, i.e., an MFT group focused on ABSR.

Selection criteria were: adolescent aged between 12 and 18 years, whose psychiatrist determined that they met Berg's criteria for school refusal [7] and had a DSM-5 anxiety disorder as primary diagnosis, whose period of school absenteeism range from 2 consecutive weeks through 18 months, and who had attended mainstream schools and had not presented any learning difficulties before school refusal began.

The study was offered to all families who participated in a MULTI FAST group in 2022 and 2023. From 4 to 12 weeks before planned MFT started, we met participants at an appointment, informed them about the study, and invited them to take part in it.

## Data collection

Around six months after the end of MFT, each parent included in the study was contacted to arrange a semi structured interview planned to last about one hour. C. Snegaroff (CS) conducted the interviews alone at Maison de Solenn-Maison des Adolescents, where she works as a psychologist. She was not present at the MFT and did not know the parents. The interview guide was developed by the Maison des Adolescents research group at Cochin Hospital, after the group reviewed the international literature on MFT and ABSR.

All researchers have undergone specific training in qualitative methods. CH, CB, ML, SS, AH and MRM have extensive experience in qualitative research and have already published several papers that use qualitative methods. The main points explored in the interview guide are the parents' experience of the multifamily group, and their experiences in relation to the objectives of the therapy. The interview guide is presented in the appendix. Each interview was audio recorded and transcribed with the permission of the participants. Sampling was consecutive, given that all participants in the MULTI FAST groups in 2022 and 2023 were asked to participate in this study. The recruitment period for this study started on May 11, 2022, and ended on November 23, 2023.

## Data analysis

A phenomenological research design was employed to explore the experiences of parents who participated in our MULTI FAST groups. Phenomenology is a nonprescriptive approach to research that allows the essence of experience to emerge, yet anchors data analysis in the participants' unique representations [47]. The aim is to explore personal experience and the subjective perception of an object or event. Our research approach is phenomenological in that it involves detailed examination of the participants' personal perceptions and lived experiences. To analyze our interview data, we used the *Interpretative Phenomenological Analysis (IPA)* method [48,49], a well-established qualitative methodology used to explore in depth how individuals perceive particular situations they are facing and how they make sense of their personal and social world. Applying IPA, we conducted an in-depth qualitative analysis, beginning with a detailed case-by-case study of each interview transcript by an iterative inductive process. Specifically, to obtain a holistic perspective, we conducted several close detailed readings of each interview, noting points of interest and significance. Step by step, we proceeded to describe the analytic themes and their interconnections, while taking care to preserve a link back to the original account. IPA thus involves navigating between different levels of interpretation [48]. The last stage involves the production of a coherent ordered table of the themes [49]. The procedure for data analysis was inductive; data from the

literature were analyzed secondarily. The data collection and analysis took place at the same time. In this type of study, sampling, continued data collection and the development of themes and sub themes are interdependent and simultaneous. The analysis influences the data collection by leading us to redefine the research question, to find counter-examples, and to investigate new pathways [50].

### Validity

To insure the validity of our qualitative research, we compared the researchers' coding. Two trained researchers (CS, AH) independently coded and interpreted all of the parent interview data. The two coders discussed the emerging codes in repeated meetings with other members of the research team (EH, SG, SM, ML, CH, CB), all of whom had read the transcripts. All the researchers are women except SM who is a man. These discussions helped to identify potential themes in the data that might not yet have been captured by the codes and enabled us to clarify or modify the coding to increase the consistency and coherence of the analysis by ensuring that the themes we identified reflected the data accurately and that the analysis was not confined to one perspective. Multiple discussions allowed us to eliminate systematic differences due to variations in interpretation. Validity was also enhanced by the care we took to distinguish clearly between what respondents said and how we interpreted it or took it into account [49].

Member-checking (also known as informant feedback or respondent validation) was practiced, for it is a vital way for interpretive researchers to verify the trustworthiness of their research [51]. When the qualitative analysis of the parents' data was completed, a summary of the thematic results was mailed to the parents, who were asked to provide feedback, reactions, and comments. Participants were also asked to share these preliminary results with their partners who had been unable to take part in the interview. Eighteen of the 31 parents provided written or verbal feedback, which was incorporated in the final results. This methodological aspect of the study provided a source of testimonial validity for the qualitative results; it also enabled us to take participant feedback about their interpretation into account [52] and to assess the degree to which the themes resonated with the parents' experiences.

### Ethical aspects

Parents were informed of the aims of the study and of the voluntary nature of their participation, and that all their answers would remain confidential. Participants provided informed verbal consent. A copy of the information document was given to them prior to their participation in the research. The information given to the adolescents and parents was recorded in the adolescent's medical file, and the non-opposition of the adolescent and his/her parents was recorded by the investigator. The ethics committee approved the study on May 9, 2022 (Approval Number: 21.04640.000059).

## Results

Thirty-seven parents were asked to participate in this study, and 31 agreed (84%). The 6 who refused (or did not respond) either pointed out their lack of time or gave no explanation.

### Participants' characteristics

Of the 31 parents, 19 were mothers and 12 fathers. Parents'ages ranged from 41 to 65 years. The mean age was 51.5 (SD = 6.45) for fathers and 48.7 (SD = 4.77) for mothers. The country of birth was France for 71% of parents; 19% of parents were divorced and 6% single or not living with a partner. Of the 31 parent participants, 5 (16.5%) were teachers and 5 more hourly workers, 4 (13%) unemployed and 4 others sales workers, 3 (10%) artists and 3 business owners, 2 (6%) engineers and 2 nurses, one retiree, one doctor, and one civil servant. Adolescents were aged from 13 to 17 years old; 48% were girls and 52% boys. Duration of school refusal was 5–16 months. In our population, the minimum number of school days missed was 92 days.

Iterative thematic analysis enabled us to develop a conceptual map (Fig 1) comprising four domains: (1) Changes in parents' representations of their adolescent; (2) Parents looking at themselves in a new light; (3) Parents expressing their distress to their adolescent; (4) New interactions. We go on to describe each domain in the four sections that follow. Each section is organized in the same way: the definition of the domain; its division into underlying themes and subthemes; and support of these constructs through representative quotations.

Data saturation was reached with the analysis of the 31 interviews, that is, the point at which in-depth analysis of the interviews no longer results in the emergence of new themes [53]. Data analysis becomes redundant and no longer provides new elements.

## Changes in parents' representations of their adolescents

**Parents feel they better understand their children's distress and needs.** Parents reported that hearing other teens recounting their experiences and expressing their difficulties gave them access to their child's experience and distress (Q1). The demand for greater autonomy and less pressure from parents, so often expressed by teenagers, is better heard and accepted when voiced by someone other than one's own child (Q2).

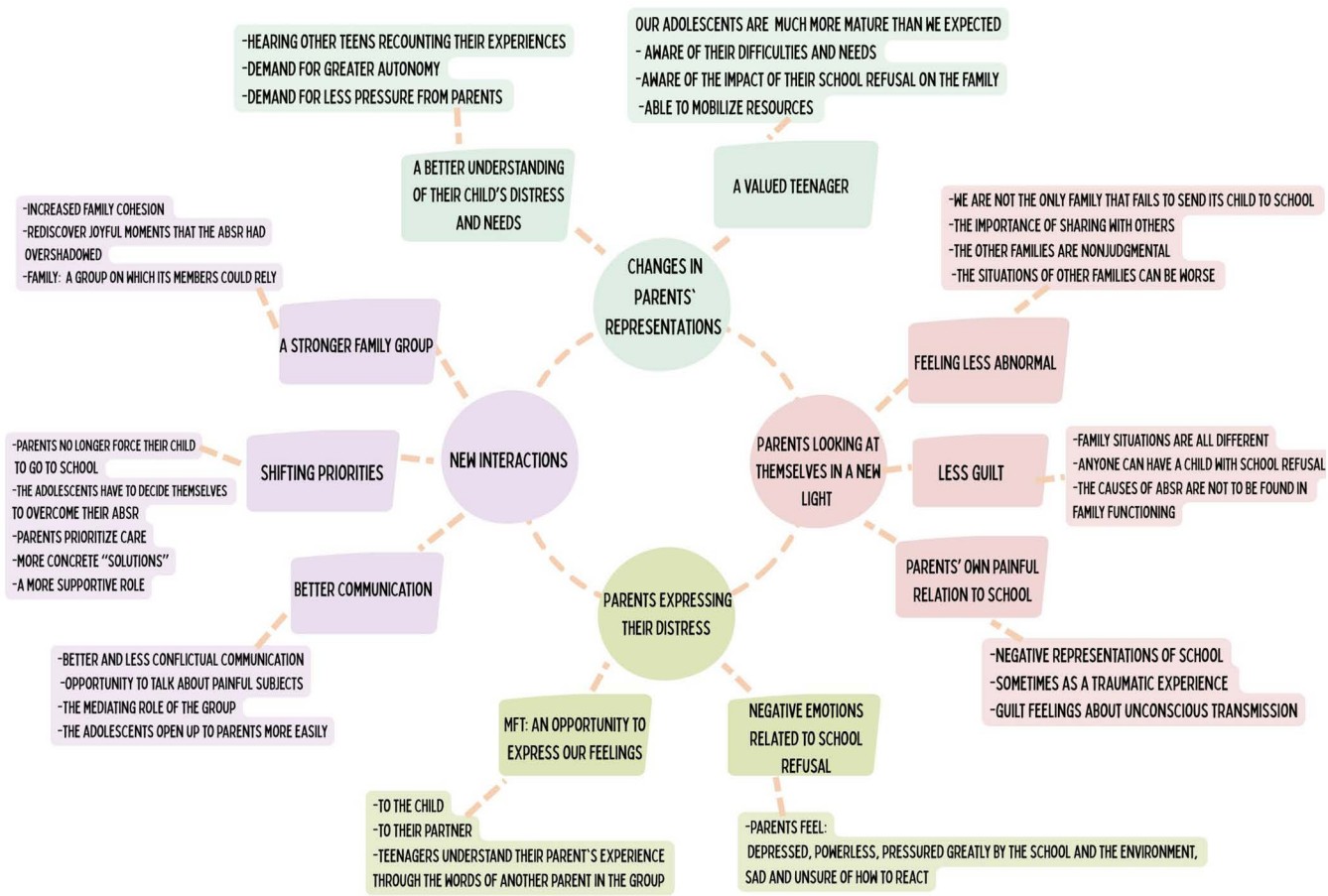

**Fig 1. Concept map: domains and themes of parents' perspectives on multifamily therapy for ABSR.**

*Q1: Father 1: "It was incredible, it put into words things that [she] couldn't put into words, and that's huge too. For me, it was exactly that. The anxiety that sets in before leaving for school, the 'I'm going to make it but I can't, he really put it into words."*

*Q2: Mother 18: "... that we let go, it came up a lot with them, it was an almost unanimous request."*

**A valued teenager.** Most parents stressed that MFT has changed how they viewed their child, whom they found to be much more mature than they expected. Their teenagers were aware of their difficulties and needs (Q3) and of the impact of their school refusal on the family (Q4), but also able to mobilize resources that parents feared had disappeared and, sometimes, to demonstrate ability to change during therapy.

*Q3: Mother 8: "I thought they were impressively mature... they expressed themselves in a very clear-sighted way."*

*Q4: Mother 15: "I also learned that he was very worried about me, about his father too, about our life... the impact he had on us."*

**Parents looking at themselves in a new light**

**Feeling less abnormal.** Recognizing themselves in the experience of other families reassured parents who could thus feel that they were no longer the only family that fails to send its child to school (Q1). Many parents underlined the importance for them of being able to share with others whom they could perceive as understanding and nonjudgmental (Q2). Very often, they also perceived the situations of other families and teenagers as worse than their own, which seemed to comfort them.

*Q1: Mother 8: "... we realize that we've been confronted with similar situations, and so it creates an affinity and relief. It doesn't just affect us, and it's shared in much the same way."*

*Q2: Mother 7: "The other families didn't have these kinds of thoughts: 'They're not doing the right thing, they're not doing enough, they're doing too much."*

**Less guilt.** All parents came to MFT feeling guilty and responsible for their teenager's ABSR. Yet within the MFT group, family situations, individual backgrounds and socio-economic backgrounds were all different. Parents often concluded from the heterogeneity of the group that the causes of ABSR are not to be found in family functioning, especially when they compared themselves to parents they perceive as more competent (Q3).

*Q3: Mother 8: "To also see that we came from very different backgrounds... and that, despite all our differences, we were confronted with somewhat similar problems... In other words, it's no longer this or that person who's done wrong."*

**Parents' own painful relation to school.** In each group, at least one parent (a total of 5) recounted how they became aware, during the therapy, of their own painful relationship with school, sometimes as a traumatic experience (Q4). MFT includes several exercises on the participants' (parents' and teenagers') images of school. For example, we asked them to give the first three words that came to mind to describe school, or to recount a good and a bad memory of school, or to draw a picture of their school when they were teenagers. Parents were often surprised by how they talked about or represented school, because they didn't think they had any negative representations of school. This awareness sometimes led to guilt feelings about an unconscious transmission (Q5).

*Q4 Father 22: "Well, there are things that come up, things that are very deep down, things where you say to yourself: 'well, that's part of my schooling too'."*

*Q5 Mother 15: "Are there things that we went through as children that, um... stresses that we passed on to our child?".*

### Parents expressing their distress

**Negative emotions related to school refusal.** Parents insisted that their role as parents was to enable their children to go to school, get an education and then live their adult lives without them. When their teenager was no longer able to attend school, parents were overcome by very negative emotions. Parents with children distressed by school refusal felt depressed, powerless, and pressured greatly by their child's school and the environment. They experienced moments of sadness and despair (Q1). They felt unsure of how to react (Q2).

*Q1 Mother 17: "... a situation that is often hard, hard, hard, depressing, dark, sad."*

*Q2 Mother 10: " We try a bit of everything, to get angry, to speak nicely, it doesn't work, I think we're all a bit helpless in the face of this."*

**MFT: An opportunity to express feelings.** MFT offered parents an opportunity to express their discomfort or distress to their child, and/or partner (Q3). Some parents did not immediately express their feelings, but other parents did. In this way, the teenagers understood their parent's experience through the words of another parent in the group. Other parents, expressing their difficulties, could take on the role of spokesperson (Q4).

*Q3 Mother 15: "We each had to choose a photo and … I immediately saw a burning house... I sometimes have the impression that there's nothing left to hold on to."*

*Q4 Mother 4: "... my son also saw that other moms, other parents, could ask themselves the same questions... I think that helped."*

### New interactions

**Better communication.** The group's presence encouraged communication. Communication within the family was described as less conflictual during MFT (Q1). Parents insisted on the importance of the group: it was the group's presence that made it possible to talk about difficult and painful subjects, without this leading to conflict, as it was before when the same subjects were discussed only within the family. Parents emphasized the group's mediating role.

*Q1 Father 5: "It was a moment when we knew we were going to talk about it in a non-confrontational way, because we were in a group. That kind of cuts out the personal reactions that can lead to rejection or closure."*

In these interviews after the MFT ended, several families reported better and less conflictual communication, particularly about the issue of school refusal itself (Q2). Some parents feel that their teenagers open up to them more easily, sometimes to reassure them (Q3).

*Q2: Mother 15: "It's much less of a taboo subject; in the family we talk about it a lot more."*

*Q3: Mother 8: "I think he's understood that the fact of not speaking to us was a factor of growing concern."*

**Shifting priorities.** A few months after the end of the therapy, most parents reported that the adolescents had to decide themselves to overcome their ABSR (Q4). For some, the exercise of parental authority had shifted from the issue of schooling to that of the duty of care. In other words, parents were no longer forcing them to go to school (which didn't work) but were making them to go and see their psychologist, for example, because they were now prioritizing care. Parents nonetheless expressed questions and doubts. They wonder whether their adolescent's reactions were related to adolescence or anxiety. Moreover, some still felt lost and would have liked more concrete "solutions" to help them cope with difficult situations (Q5).

*Q4: Mother 18: "I think therapy has a lot to do with letting go and not substituting yourself for your children".*

*Q5 Father 5: "My biggest regret is that there was no coaching. Through the various exercises, even if we learned something, we weren't given the solution."*

Parents also felt that they were now playing a more supportive role (Q6). Simply participating in MFT could be a way for parents to show their child that they were involved alongside them.

*Q6: Mother 8: "It's not a question of knowing everything, but of knowing how to accompany."*

**A stronger family group.** Many parents described increased family cohesion during the MFT sessions, promoted by activities devoted to family functioning (Q7), during which family members sometimes reconnected with the joyful moments that the school refusal issue had overshadowed, but also by the simple fact that they came to therapy and spent a few hours together. In some cases, the family eventually came to be seen as a group on which its members could rely (Q8).

*Q7: Mother 8: "All the activities, where we had to define the family and everyone's role, using images, metaphors, trying to find things that represented us, that was a real factor of closeness and consideration"*

Q8: *Mother 21: "It reinforces the notion of... of family, of the family's ability to cope with difficulties."*

**Respondent validation/participant feedback.** Eighteen participants provided feedback. All reported that that they agreed with the themes that had emerged from the analysis. More than half had kept in touch with other parents after the therapy had ended.

## Discussion

The aim of this study was to describe the experiences of parents who participated in an MFT program about their adolescent's school refusal. The phenomenological qualitative analysis of the 31 interviews highlighted several results. The first was the importance of the group: parents emphasized that being with other families facing the same problem as they did — their adolescent's school refusal — made them feel less alone, less stigmatized, less judged, and less guilty. It thus enabled them to regain confidence in themselves and their own abilities. The results also showed that a child's school refusal led to a deep sense of parental distress, sadness, and helplessness. The MFT program allowed them to express this to their teenager and the rest of the family. Some parents discovered during the MFT program that they had a negative vision of school is negative, i.e., they realized that as children or teenagers their school experience had been difficult and that this had affected how they talk about school with their teenager. Adolescents also discovered how their parents felt at school when they were pupils and were sometimes surprised by what they saw. Another important result was the change in how parents looked at their teens. Rather than seeing them as failures, they realized in the MULTI FAST group that their children have important skills and can and do analyze what's going on for themselves and find solutions. They

also understood better what their teenager needs, and how their reaction as parents can help or, on the contrary, increase the teen's anxiety. Finally, parents described how participation in a MULTI FAST group led to changes in family interactions and communication and strengthened family bonds.

### Improving parental self-efficacy

The results of the parent interview analyses reveal a high level of parental distress. As we saw in the interviews, parents described feeling judged, guilty, and powerless to help their child. The results also show that the MFT program enabled parents to regain their self-confidence and feel less guilty. They once again felt able to help their teenager without putting pressure on him/her, and to be part of the solution. Carless et al. studied the level of parental self-efficacy in these ABSR situations and showed that the feeling of parental competence is an important prognostic factor [54]. One of the aims of the MULTI FAST program is to increase parents' sense of competence and self-efficacy in parenting.

Our research shows that encounters with other parents and family systems help to restore parents' self-image. Parents recognize that they have something in common with other parents and feel part of the parent group, thanks to the identification and counter-identification movements that unfold in MFT. When mothers and fathers recognize themselves in the discourse or behavior of other parents, they can escape both their feeling of abnormality ("I'm not the only one") and their experience of isolation ("I'm not alone anymore"), by being able to confide in others without feeling judged.

The heterogeneity of the participants also proved to be a reassuring factor, as the differences between the backgrounds of the families and adolescents were very often perceived to their advantage by the parents, who generally felt that they were in a better position than others, in terms of the seriousness of their child's condition. Moreover, the wide diversity of family profiles, particularly in terms of socioeconomic status, often led parents to conclude that family functioning could not be the cause of school-based anxiety, especially when they perceive other families as more competent than themselves. This helped to alleviate their frequently expressed guilt feelings. These results also show that this therapy helped parents to perceive their teenager as the key player in overcoming ABSR — another essential element helping to relieve the pressure felt by parents.

Parents' painful experiences and emotions are also the topic of conversations between families. According to one of the main organizing principles of MFT, the aim is for parents to become each other's "co-therapists", with the system sending them "a message of confidence in their ability to resolve their problems themselves" [46]. Attending MFT sessions with their teenager can thus be perceived as a way of escaping passivity, eventually being able to act and thus feel empowered as a parent. Ausloos developed this fundamental notion in systemic family therapy: one of the main objectives is for families to regain confidence in their own ability to get through difficulties, rather than waiting for solutions to come from outside [55].

### Mentalization and MFT

Our results show that these parents felt they had a better understanding of their child's distress and anxiety during and after MFT. Most said that their children talked to them more freely during these sessions, without fear of triggering conflict. But parents also reported that listening and talking to other teenagers was a key experience that helped them to realize what their own teenager was going through and gave them access to his/her distress. Demand for greater autonomy and less pressure from parents, so often expressed by teenagers, is also better accepted when voiced by someone else's child.

At the same time, the image of helpless children was being replaced by one still more valued: clear-sighted adolescents — capable of thinking for themselves and asserting their desires, particularly for autonomy, aware of the impact of their difficulties on their family and able to marshal resources. For example, some activities during MFT invite adolescents to express themselves in front of the whole group, thus putting some of them face to face with their anxiety. Some parents cited seeing adolescents — their own or others — overcome their difficulties and talk in front of the group as a defining

moment of the therapy. As students with school refusal have low levels of self-esteem [56], the impact of these new parental representations on their adolescent should be further explored.

Mentalization also worked for teenagers, helping them to better understand their parents and their reactions. Parents could also seize the opportunity to express their feelings to their children during activities involving the whole family. Some also felt that by listening to other parents' experience, their child gained access to their own feelings and thoughts and was thus better able to understand them.

Finally parents' own experiences of school are also explored during MFT. Results show that among parents who agreed to be interviewed, at least 7 (one in group 3 and two in groups 1, 2 and 4) became aware during the therapy of their own painful relationship with school. Some realized for the first time how traumatic this experience had been.

For both parents and teenagers, expressing this painful relationship with school can help to better understand the parent's reactions to the adolescent's ABSR. Behind the telescoping of these experiences lies the question of the transmission from parent to child of their anxiety about the school system [57] and the associated guilt parents also expressed. Gallé-Tessoneau and Dahéron described among children in their practice with school refusal, "a history of painful experiences on the part of a parent or grandparent concerning schooling, or even traumatic experiences that hinder the child's vision of school as a place of security." [58].

### Impact on family interactions

Families faced with an adolescent's ABSR must deal with a very difficult situation: their adolescent has been unable to attend school, sometimes for several months. Improving family resilience, that is, its ability to deal with challenges and adversity [59] is therefore essential. The results show that family resilience was enhanced by encounters with other families facing the same problem and by inter family support.

The importance of working on family communication is another key point of the MFT program: family communication involves sharing information, ideas, thoughts, and emotions among family members [59,60]. In the third theme to emerge from the phenomenological analysis of parent interviews, parents described new interactions with better communication and a stronger family group. Some of the activities proposed during MFT sessions are designed to work on family communication. For example, during certain MFT sessions, we ask families to describe recurring blockages or conflicts within the family, and then to reflect on strategies for reacting and managing them differently. This requires them to work on family functioning, which involves the family members' ability to collaborate [59].

Finally family emotional expressiveness refers to the prevailing mode of emotional expression within the family context, encompassing both nonverbal and verbal cues [61]. For parents, successfully expressing their emotions within the MFT program in front of their family and also hearing their adolescent express his or her emotions were key moments in the program.

The study by Sabah and Alduais demonstrates the mediating role of the parent-adolescent relationship between family emotional expressiveness and adolescent mental health [62]. This important point is in line with the objectives of MFT: to work on the parent adolescent relationship to improve adolescent mental health, but also to work on parents' expression of positive emotions toward their adolescent.

These points show the particular relevance to our MFT program of Olson's Circumplex Model of Marital and Family Systems [63], which emphasizes three primary aspects within family systems: cohesion, flexibility, and communication.

An improved family emotional climate has been highlighted in several studies evaluating MFT for other disorders, particularly eating disorders [35,64]. The same conclusion can be drawn from our results: a few months after the end of the MFT, many of the parents who agreed to participate in our study said that communication has improved within the family. They also described fewer conflicting relationships, and these focused mostly on the adolescent's continued refusal to go to school.

Family cohesion appears to have increased during MFT. Families reported spending hours together discussing family functioning or simply reconnecting with the joyful moments that the school refusal issue had overshadowed. The impact

continued for at least several months after the therapy ended and reached the point where family could be viewed as a resource.

Most parents interviewed described a new parental position toward their adolescent. They no longer tried to force school attendance, trying instead to play a more supportive role. Often lacking the means to act on the educational front, they appeared to redefine their parental role by asserting their parental authority over the need for care and the non-negotiable presence of their teenager in the institutions or with the professionals in question.

Everyone's place in the family appears to have changed. When adolescence is doing well and school refusal does not continue, teenagers go through the second phase of the individuation separation process described by Blos [65]. Spending time with groups of people their own age, the adolescents continue their development while distancing themselves from their parents. School is an ideal place to meet people of the same age. School refusal, on the other hand, seems to freeze the adolescence process. In the period of life when teens are creating new ties outside the home, separating from their parents, and becoming independent, ABSR blocks this process. The adolescent stays at home and no longer builds new relationships.

It seems to us that MFT helps to relaunch this process. Its aim is to reinsert these children back into their place as teenagers, building relationships with people their own age and able to stand on their own two feet. When parents step aside and tell their child that he or she is the only one capable of finding a way out of this situation, they are, we propose, enabling and encouraging their adolescent offspring to "appropriate" their own history and paving the way for a psychic separation: both this appropriation and separation are essential processes of adolescence.

### Have the teenagers gone back to school?

During the interview with parents six months after the end of MFT, the parents were asked about their teenager and whether he or she had returned to school. This same question was asked in a telephone call twelve months after the end of MFT. Of the 27 adolescents who participated in MULTI FAST groups in 2022 and 2023. 15 had returned to regular school, 5 more had returned to a school with a small class size and continued psychological care, 3 had started training to learn a trade, and 4 had neither returned to school nor started a professional project.

### Limitations and prospects

Because we chose a purely qualitative method, we cannot demonstrate the therapeutic benefit of MFT for adolescents with ABSR and their parents. For example, the presence of other factors, such as treatment intensification, may play a role in the experience of the parents we interviewed. This study thus successfully explores parents' experiences of MFT, but it cannot quantitatively demonstrate its therapeutic benefit in ABSR in adolescents. The relief described by the parents may also be linked to the fact that they were able to access treatment in an adolescent psychiatry department. While France currently has a very high demand for care for adolescents with mental health problems, it has very few adolescent psychiatric departments. In the future, we plan to set up a quantitative study to assess the effectiveness of MFT for adolescents with ABSR and their parents. This quantitative study will complement the qualitative approach presented in this paper.

The results of the analysis of the interviews with the adolescents will be published soon, but the initial analyses confirm the value of MFT from the adolescents' perspective as well as their parents'. The point of view of siblings is also very important, and their presence is fundamental in MFT groups. Research specifically focused on siblings would thus be useful.

### Theoretical implications

This study sheds light on the importance of including both family and group therapy to help adolescents with ABSR. MFT allows us to work on intra family communication and the expression of emotions within the family, so that they can better understand ABSR and at the same time benefit from the help of the group, so that families can support each other as they face the same problem.

### Practical implications

It would be helpful to systematically offer families programs where they can meet other families facing the same problem. A partnership between schools and health care services is essential to provide the best possible support for these families.

### Conclusions

MFT is a worthwhile and promising therapeutic approach for parents of teenagers with anxiety-based school refusal. It differs from a parents' support or discussion group especially in that the adolescents are present. The parents hear their child and other teenagers give their point of view, as experts in this form of school refusal. Moreover, these teens are exposed to their parents' perspectives and further given the opportunity to be exposed to other parental perspectives and, of course, to spend time with their peers.

Working with the family can improve the prognosis of anxiety-based school refusal. This study clarifies why family support is so fundamental: when teens find themselves in a bad place, stop attending or even drop out of school (scenarios that can drag on for a long time), parents need help to be able to hold on, to support their teenager, and to not develop an anxious and/or depressive disorder themselves. Otherwise, the risk is that the whole family feels stigmatized by school and society and becomes isolated. Daily life at home then becomes a place of conflict, misunderstanding, and discouragement, and the severity and duration of this school refusal increase. For these reasons, let's not forget parents as the first partners in care!

### Supporting information

**S1 File. Parent interview guide.**
(DOCX)

**S2 File. ISSM COREQ Checklist.**
(PDF)

### Acknowledgments

We thank the families and all the healthcare professionals, who assisted in the successful conduct of this study. We also thank Jo Ann Cahn who translated this manuscript and Axelle Dubrez who created the concept map.

### Author contributions

**Conceptualization:** Aurélie Harf.

**Formal analysis:** Claire Snegaroff, Maude Ludot-Grégoire, Sara Skandrani, Salomé Grandclerc, Sevan Minassian, Eloïse Hellier, Aurélie Harf.

**Methodology:** Maude Ludot-Grégoire, Caroline Barry, Christine Hassler, Aurélie Harf.

**Supervision:** Sara Skandrani, Marie Rose Moro, Aurélie Harf.

**Writing – original draft:** Claire Snegaroff, Aurélie Harf.

**Writing – review & editing:** Maude Ludot-Grégoire, Rahmeth Radjack.

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
