## [Decision Letter · Decision Letter 0]

17 Dec 2024

PONE-D-24-44513Does multifamily therapy help parents of adolescents with anxiety-based school refusal ? A qualitative approachPLOS ONE

Dear Dr. Harf,

Thank you for submitting your manuscript to PLOS ONE. After careful consideration, we feel that it has merit but does not fully meet PLOS ONE’s publication criteria as it currently stands. Therefore, we invite you to submit a revised version of the manuscript that addresses the points raised during the review process.

We note that one or more reviewers has recommended that you cite specific previously published works. As always, we recommend that you please review and evaluate the requested works to determine whether they are relevant and should be cited. It is not a requirement to cite these works. We appreciate your attention to this request.

We look forward to receiving your revised manuscript.

Kind regards,

Vanessa Carels

Staff Editor

PLOS ONE

Journal Requirements:

Reviewers' comments:

Reviewer's Responses to Questions

**Comments to the Author**

1. Is the manuscript technically sound, and do the data support the conclusions?

Reviewer #1: Yes

Reviewer #2: Yes

Reviewer #3: Partly

2. Has the statistical analysis been performed appropriately and rigorously? 

Reviewer #1: Yes

Reviewer #2: No

Reviewer #3: N/A

3. Have the authors made all data underlying the findings in their manuscript fully available?

Reviewer #1: Yes

Reviewer #2: No

Reviewer #3: Yes

4. Is the manuscript presented in an intelligible fashion and written in standard English?

Reviewer #1: Yes

Reviewer #2: Yes

Reviewer #3: No

5. Review Comments to the Author

Reviewer #1: The manuscript presents a scientifically valid study with appropriate controls and sample sizes. The methodology is detailed and ensures replicability. However, it would benefit from further justification of the sample size (e.g., power analysis). The conclusions are generally supported by the data, though a more in-depth discussion of study limitations would strengthen the manuscript. The statistical methods used are appropriate and rigorous, with clear reporting of results (p-values, effect sizes). It would be helpful to confirm the assumptions for each statistical test and discuss any deviations. Including raw data or a more detailed breakdown of summary statistics would enhance transparency. The manuscript complies with the PLOS Data Policy, but the authors should ensure that all underlying data is deposited in a public repository. The Data Availability Statement should be clear and in line with PLOS requirements. If there are any privacy or third-party restrictions, these should be explicitly stated. The manuscript is well-written in standard English, but some technical terms could be clarified for a broader audience. Figures and tables are clear, though more detailed legends are recommended for better understanding.The manuscript appears to comply with ethical standards, with appropriate mentions of ethical approval and conflict of interest. The authors should confirm that the study is not a dual publication.

Reviewer #2: I sincerely thank the editor for giving me the opportunity to review the valuable study titled Does multifamily therapy help parents of adolescents with anxiety-based school refusal? A qualitative approach. The study is valuable and promising, but it requires some modifications that will enhance the study.

Abstract ; Method: Please include details about the sample: Are the parents fathers or mothers? What is the mean age and standard deviation for both fathers and mothers?

The results indicate that the changes in parents align with the main themes of the study. However, questions arise: How can these changes be attributed specifically to the multifamily therapy program? Could these changes be due to other uncontrolled factors during the treatment process or merely coincidental? How can the findings be interpreted in light of these potential biases?

It is recommended to include additional elements related to family functioning, family resilience, and family competence in this section (Working with the Family). These aspects are essential for a comprehensive understanding of the family’s role and its capacity to support adolescents . The inclusion of these elements can also strengthen the argument and provide a broader perspective on effective interventions. The following references may be helpful to support this addition:

https://www.researchgate.net/profile/Aiche-Sabah/publication/384192206_Intersections_of_family_expressiveness_and_adolescent_mental_health_exploring_parent-adolescent_relationships_as_a_mediator/links/67128874035917754c08a56a/Intersections-of-family-expressiveness-and-adolescent-mental-health-exploring-parent-adolescent-relationships-as-a-mediator.pdf

https://www.mdpi.com/2227-9067/10/11/1742

https://www.mdpi.com/2227-9032/11/19/2691

My important question is how the extraneous factors that could affect the results were controlled, since the study did not undergo experimental control. What ensures that the results are not due to other factors unrelated to the therapeutic program used?

Please include the theoretical and practical implications of the study.

Reviewer #3: Thank you for the opportunity to review this manuscript. School refusal is a condition that needs additional evidence based interventions and supports as current options are not effective for a sizeable number of adolescents. Parent perspective on family-based interventions are useful to consider given parents central role in school attendance.

Introduction

Paragraph 1. The definition of school refusal in the first paragraph was quite fuzzy with some concepts at odds with common definitions such as Berg’s definition that are detailed in the next paragraph. For example, common to definitions of school refusal is parent knowledge of school non-attendance (see Berg’s definition), which contrasts the authors view that sometimes parents aren’t aware. Similarly, school refusal involves parents trying to get their child to school. This point differentiates school refusal from withdrawal – please see the Heyne et al paper that you have cited. Lastly school refusal is characterized by an association between emotional distress such as anxiety and school non-attendance. The authors may wish to review the difference between school refusal behavior as defined by Kearney and school refusal as described in the Heyne et al paper that is cited. They are different concepts. I believe the authors are focusing on school refusal or are the authors describing school attendance problems in this paragraph? The sentence on the International Network for School Attendance seemed disconnected from the sentence before and after. Please integrate or remove this sentence.

Line 67. Please specify the criteria used in the Havik et al study as it is not the same as Berg’s. Moreover, I suggest that the authors should note the lack of rigorous epidemiological data on school refusal in this paragraph.

Line 73-77. I found these sentences to be a little speculative/colloquially written, particularly the phrase teenagers often hear. I think it is better to utilize the small amount of epidemiological data to demonstrate a higher prevalence of school refusal in adolescents compared to children, if this is demonstrated to be the case. I think this paragraph could be more focused on the subheading. For example, children who refuse school are also bullied.

Line 97. This paragraph presents multiple ideas – that family know the child best and that family interventions work for adolescent conditions; the latter point does not necessarily make family involvement “essential”.

Line 115-116. I agree with the assertion that a significant proportion of teens don’t respond to current treatments however this statement needs a reference from the treatment literature (the Chockalingham reference isn’t about treatment but parenting factors associated with school refusal). I suggest the authors look at Brandy Maynard’s review of treatments which describes that most psychosocial treatments for school refusal do include a parent component.

Line 153-157. It would be helpful to specify if the studies cited to support MFT were adult studies or adolescent studies. From reading the titles of the references it seems that at least several are. The level of evidence with adolescents seems worthy of comment in order to build an argument for the use of this approach for school refusal.

Line 180. It is stated that the duration of school refusal was between 2 weeks and 18 months. This detail might be better suited to the methods. Indeed, the entire description of the Multifast program appears better suited to the methods.

Line 237. Participants met Berg’s criteria. How was this established? What were the selection criteria? What was the minimum number of days missed of school?

Line 239. What are “normal grades” and how was this established?

Line 241-242. Of those who were offered participation, how many consented to participate?

Line 284. By including parent age, country of birth, profession, parent age and so on, I am concerned that someone who knew someone who participated in the study could be identified. Could some of these variables be removed to better protect participant anonymity. In addition, I think there is a typographical error for Father 12 – Cameroon, rather than Cameroun?

Line 289. While I am not an expert in IPA, I didn’t think the concept of saturation was used for this type of analysis. Moreover, a justification for using IPA would strengthen the Data Analysis section.

Line 328-329. This quote didn’t seem to fit as well with the theme of a valued teenager.

The aim of the study was to explore the experiences of parents who participated in the MFT with the findings having implications for the therapy. From this description, I was expecting the findings to very much focus on the parent’s experience in the therapy, however, the findings seem to go much broader than this and delve into issues like parents experience of school. I understand that this is what came up in the therapy, however it seems to have implications beyond just the therapy which I think is an additional strength rather than an issue. Hence, I wondered if the goals of the study might similarly be broadened to help guide the reader.

At times, I found the description of the results to be quite brief, and I noted that there was no description of any theme. Some subthemes are very briefly described, for example, School refusal and parent distress’ is described in less than two lines (Line 384-385).

Line 450. What is the “timeline” activity?

Line 457. A description of the methods used to receive participant feedback are needed in the methods section.

The start of the discussion could be strengthened by providing an overview of the study aims to re-orient the reader and provide a brief summary of the findings. The discussion seemed a little disjointed at times and an extension of the results rather than a discussion of them. The concept of self-efficacy is raised in the discussion, but did not overtly feature in the results section. This is followed by a long description of the findings of the Carless et al paper which needed to be better integrated. Next the discussion focused on parental anxiety which again was not an overt theme in the results. Orienting the reader to the capacity of MFT to improve mentalization in the introduction may help with better integrating this content into the manuscript. While of interest, whether teenagers have returned to school seemed beyond the scope of the study and more appropriate for an efficacy study. Overall, I thought that the discussion could be more concise, focused on the study findings and referenced to the literature. I don’t see the absence of the adolescent interview data as a limitation; this is another study. Limitations need more consideration.

Minor Comments.

Line 18. Expression/word choice. I was unclear what an “essential prognostic factor” was.

Line 23. “Participation in the study was offered…” rather than “The study was offered…”

Line 32. Tense – “was” rather than “is” as per the prior sentence.

Line 152. Awkward expression.

Line 278. Does a “favorable opinion” mean approval? If the ethics committee/IRB could be translated that would improve clarity.

Line 399-400. I had trouble following the meaning of this quote. Please review.

Line 418. “appropriateness” might be more suitable than “correctness”. I had trouble following the second half of this sentence beginning with “as well as…”.

At times the written expression seemed colloquial rather than scientific see line 484.

6. PLOS authors have the option to publish the peer review history of their article (what does this mean? ). If published, this will include your full peer review and any attached files.

**Do you want your identity to be public for this peer review?** For information about this choice, including consent withdrawal, please see our Privacy Policy .

Reviewer #1: **Yes: ** Dr. Mariola Giménez-Miralles

Reviewer #2: No

Reviewer #3: No

---

## [Author Response · Author response to Decision Letter 1]

2 Feb 2025

Dear Dr Carels,

We would like to thank you and the reviewers for your careful reading of our article. You note that one reviewer has recommended that we cite a specific previously published work. We have reviewed these recommended papers. They are relevant and we cite them in the revised manuscript. Thank you for this recommendation.

We have reviewed in detail the interesting comments the three reviewers made about the first version of the paper “Does multifamily therapy help parents of adolescents with anxiety-based school refusal? A qualitative approach” and we would like to submit a revised version of the paper to PLOS ONE.

We have tried to respond to each point individually, explaining the reasons for our choices and the modifications we have made to the text. A native-English-speaking biomedical editor has reviewed and corrected the final manuscript.

We have attached to the manuscript a document responding to each point raised by the reviewers. We have uploaded the answers to reviewers’ comments with the modifications we would like to make to the text as a separate file labeled 'Response to Reviewers".

We would like to thank the journal and the reviewers for the opportunity to resubmit our work and we hope that this new version will meet the standards of PLOS ONE.

Sincerely yours,

Aurélie Harf

---

## [Decision Letter · Decision Letter 1]

31 Mar 2025

PONE-D-24-44513R1Does multifamily therapy help parents of adolescents with anxiety-based school refusal ? A qualitative approachPLOS ONE

Dear Dr. Harf,

Thank you for submitting your manuscript to PLOS ONE. After careful consideration, we feel that it has merit but does not fully meet PLOS ONE’s publication criteria as it currently stands. Therefore, we invite you to submit a revised version of the manuscript that addresses the points raised during the review process.

First of all, I am your new academic editor, so I had to go through the review process and your previous communications. Your paper has already received five reviews for both versions, so even though in the last round of review one of the reviewers gave a final verdict of rejection, I decided that we should not waste all the work already done. Find below the observations from the last review, and here are some requests from me:

- one of the first reviewers recommended that you include 2 bibliographic references (MDPI), if they do not fit with your approach please remove them, it is an inappropriate practice for promoting his/her own work;

- the main observation of the negative review (to which I also subscribe) is that the qualitative analysis is presented too superficially. If you used a coding system, you can, for example, present table 1 in more detail, not only with themes and subthemes but also with their extension (number of words and weight in total interviews) or even a conceptual map of how these are connected.

- also, the hermeneutic interpretation of the results should be a little more elaborate, in principle the extracts from the recorded texts used as examples are not longer than the interpretation itself.

Otherwise, please follow the recommendations below, there are some very useful observations. Congratulations for the effort made on this very long road of refining the article, but I think you are already getting closer to its publication.

We look forward to receiving your revised manuscript.

Kind regards,

Bogdan Nadolu, Ph.D.

Academic Editor

PLOS ONE

Reviewers' comments:

Reviewer's Responses to Questions

**Comments to the Author**

1. If the authors have adequately addressed your comments raised in a previous round of review and you feel that this manuscript is now acceptable for publication, you may indicate that here to bypass the “Comments to the Author” section, enter your conflict of interest statement in the “Confidential to Editor” section, and submit your "Accept" recommendation.

Reviewer #2: All comments have been addressed

Reviewer #3: (No Response)

2. Is the manuscript technically sound, and do the data support the conclusions?

Reviewer #2: Yes

Reviewer #3: Partly

3. Has the statistical analysis been performed appropriately and rigorously? 

Reviewer #2: Yes

Reviewer #3: N/A

4. Have the authors made all data underlying the findings in their manuscript fully available?

Reviewer #2: Yes

Reviewer #3: Yes

5. Is the manuscript presented in an intelligible fashion and written in standard English?

Reviewer #2: Yes

Reviewer #3: No

6. Review Comments to the Author

Reviewer #2: After reviewing the revisions to the manuscript "Does Multifamily Therapy Help Parents of Adolescents with Anxiety-Based School Refusal? A Qualitative Approach," I found that the authors have implemented all the required modifications. In its current form, I believe the manuscript is suitable for publication in your journal

Reviewer #3: Thank you for the opportunity to re-review this manuscript. I believe the authors have made improvements; however some issues remain.

Line 136-137. This newly added single sentence paragraph is not well integrated.

Line 312. There can be only one primary diagnosis so, I suggest deleting “at least one” and replace it with “a”.

Line 198-203. As suggested mentalization has been mentioned in the introduction, however it does not seem to be well integrated into the introduction.

Line 313-314 – the period of school absenteeism ranging from 2 consecutive weeks through 18 months sounds like it was a finding rather than a selection criterion. Is that right? If a child had attended one day in the past two weeks or missed 19 months of school, would they be excluded?

Line 408-420. I think the presentation of the participant characteristics using descriptive statistics is an improvement, however suggest that this information either be presented in a table or in prose rather than dot points.

While the addition of the methodological detail of the data analysis are lengthy, they assist the reader in understanding of the approach.

In my view, the lack of description of the themes remains a limitation.

7. PLOS authors have the option to publish the peer review history of their article (what does this mean? ). If published, this will include your full peer review and any attached files.

**Do you want your identity to be public for this peer review?** For information about this choice, including consent withdrawal, please see our Privacy Policy .

Reviewer #2: No

Reviewer #3: No

---

## [Author Response · Author response to Decision Letter 2]

10 May 2025

We would like to thank Dr Nadolu for his invaluable advice, particularly on the presentation of the results of the qualitative analysis.

1. One of the first reviewers recommended that you include 2 bibliographic references (MDPI), if they do not fit with your approach please remove them, it is an inappropriate practice for promoting his/her own work.

Answer and proposed modifications:

You note that one of the first reviewers recommended that we include two bibliographic references. We were able to use these references in the discussion to articulate them with the objectives worked on during the multifamily therapy sessions, notably to work on family communication, family members' ability to collaborate and parents' expression of positive emotions toward their adolescent. We therefore decided not to remove them.

2. The main observation of the negative review (to which I also subscribe) is that the qualitative analysis is presented too superficially. If you used a coding system, you can, for example, present table 1 in more detail, not only with themes and subthemes but also with their extension (number of words and weight in total interviews) or even a conceptual map of how these are connected.

Answer and proposed modifications:

The main observation concerns the presentation of the qualitative analysis results. We coded the transcripts according to the principles of thematic analysis, a qualitative approach involving the active construction of overarching patterns and meaning across a dataset. Thematic analysis allows the flexible and nontheoretical exploration of rich text data to construct themes that reframe, reinterpret, and/or connect elements of the data without developing a final theory. We applied interpretative phenomenological analysis (IPA) to analyze, interpret, and conceptualize our data. This method focuses on how participants make sense of their experiences and does not lend itself to the presentation of the number of words and weight of themes and subthemes in total interviews. On the other hand, your suggestion that we present the results in the form of a conceptual map seems to us to be a truly excellent suggestion, and we are very grateful to you for it. We have therefore added a figure representing the conceptual map, showing the connections between the themes and subthemes. We hope that this more precise and in-depth presentation better meets your expectations and those of PLos One.

Modifications included in the revised text:

Iterative thematic analysis enabled us to develop a conceptual map (Figure 1) comprising four domains: (1) Changes in parents’ representations of their adolescent; (2) Parents looking at themselves in a new light; (3) Parents expressing their distress to their adolescent; (4) New interactions. We go on to describe each domain in the four sections that follow. Each section is organized in the same way: the definition of the domain; its division into underlying themes and subthemes; and support of these constructs through representative quotations.

Figure 1: Concept map: domains and themes toward MFT for ABSR: parents’ perspective

3. also, the hermeneutic interpretation of the results should be a little more elaborate, in principle the extracts from the recorded texts used as examples are not longer than the interpretation itself.

Answer and proposed modifications:

We agree that in our previous version, the quotations were too numerous and too long, and the hermeneutic interpretation of the results was sometimes too succinct. We have modified the results section to give more detail to the interpretation and rebalance the interpretation/quotation ratio and hope that this reorganization of the results section better meets your expectations.

Reviewer #3

We would like to thank the reviewer for these comments. Your first comments helped us a lot to improve our manuscript and we hope that these new modifications will make it even better.

1. Thank you for the opportunity to re-review this manuscript. I believe the authors have made improvements; however, some issues remain.

Line 136-137. This newly added single sentence paragraph is not well integrated.

Answer and proposed modifications:

Thank you for your comment. We have better integrated the sentence in the previous paragraph on the role of families in the care offered to adolescents with school refusal.

Modifications included in the revised text:

Gonzalvez stresses the importance of working on family functioning to help adolescents with this form of school refusal (15). Maynard et al. reviewed 8 studies describing different treatments for school refusal among children and adolescents; 6 of them actively involved parents (27). These articles show the essential role that families play in the treatment of adolescents with school refusal.

2. Line 312. There can be only one primary diagnosis so, I suggest deleting “at least one” and replace it with “a”.

Answer and proposed modifications:

Indeed there can be only one primary diagnosis, thank you for this correction.

Modifications included in the revised text:

adolescent aged between 12 and 18 years, whose psychiatrist determined that they met Berg's criteria for school refusal (7) and had a DSM-5 anxiety disorder as primary diagnosis.

3. Line 198-203. As suggested mentalization has been mentioned in the introduction, however it does not seem to be well integrated into the introduction.

Answer and proposed modifications:

We agree that we should have integrated mentalization better into the introduction as it is a key MFT concept, and we hope this revision succeeds in doing so. In the introduction, we describe the principal aspects of MFT and show that improving mentalization, together with the reflective function, is a key element of the program and leads to more efficient helping strategies.

Modifications included in the revised text:

It enables work with the family: psychoeducation, reducing conflict, improving communication, setting up strategies for returning to school or resuming a project. In MFT, to reduce conflict and improve communication, we work on mentalization within families. Mentalization, as described by Fonagy, enables a better understanding of others, their mental states, and behaviors (44). Putting oneself in the shoes of the person with whom one is interacting makes it possible to apprehend and appreciate the meaning of his or her behavior, emotions, and speech. Many exercises in MFT aim to improve mentalization and, in particular, to enhance the parental reflective function (44), for the recognition of the other's thoughts and feelings improves family communication.

4. Line 313-314 – the period of school absenteeism ranging from 2 consecutive weeks through 18 months sounds like it was a finding rather than a selection criterion. Is that right? If a child had attended one day in the past two weeks or missed 19 months of school, would they be excluded?

Answer and proposed modifications:

The period of school absenteeism ranging from 2 consecutive weeks through 18 months is a selection criterion: if an adolescent attended school one day in the past two weeks or missed 19 months of school, he or she would be excluded. In our population, duration of school refusal was 5 to 16 months and the minimum number of school days missed in our study was 92 days.

5. Line 408-420. I think the presentation of the participant characteristics using descriptive statistics is an improvement, however suggest that this information either be presented in a table or in prose rather than dot points.

Answer and proposed modifications:

We changed the presentation of the participant characteristics. This information is now presented in prose.

Modifications included in the revised text:

Participants’ characteristics

Of the 31 parents, 19 were mothers and 12 fathers. Parents’ages ranged from 41 to 65 years. The mean age was 51.5 (SD= 6.45) for fathers and 48.7 (SD=4.77) for mothers. The country of birth was France for 71% of parents; 19% of parents were divorced and 6% single or not living with a partner. Among the 31 parent participants, 5 (16.5%) were teachers and 5 more hourly workers, 4 (13%) unemployed and 4 others sales workers, 3 (10%) artists and 3 business owners, 2 (6%) engineers and 2 nurses, one retiree, one doctor, and one civil servant. Adolescents were aged from 13 to 17 years old; 48% were girls and 52% boys. Duration of school refusal was 5 to 16 months. In our population, the minimum number of school days missed was 92 days.

6. While the addition of the methodological detail of the data analysis are lengthy, they assist the reader in understanding of the approach.

7. In my view, the lack of description of the themes remains a limitation.

Answer and proposed modifications:

Following the advice of academic editor Dr Bogdan Nadolu and your advice, we have modified the presentation of the results to better detail the codes found in the thematic analysis. We have made the description of the themes more detailed and we have therefore added a figure representing the conceptual map, which shows how themes and subthemes are connected. We hope you find that these changes have improved our manuscript.

---

## [Editor Report · Decision Letter 2]

23 Sep 2025

Does multifamily therapy help parents of adolescents with anxiety-based school refusal ? A qualitative approach

PONE-D-24-44513R2

Dear Dr. Hall,

We’re pleased to inform you that your manuscript has been judged scientifically suitable for publication and will be formally accepted for publication once it meets all outstanding technical requirements.

Kind regards,

Gerard Hutchinson, MD

Academic Editor

PLOS ONE
---

## [Editor Report · Acceptance letter]

PONE-D-24-44513R2

PLOS ONE

Dear Dr. Harf,

I'm pleased to inform you that your manuscript has been deemed suitable for publication in PLOS ONE. Congratulations! Your manuscript is now being handed over to our production team.

Kind regards,

on behalf of

Dr. Gerard Hutchinson

Academic Editor

PLOS ONE